# Identification of Insulin Resistance Biomarkers in Metabolic Syndrome Detected by UHPLC-ESI-QTOF-MS

**DOI:** 10.3390/metabo12060508

**Published:** 2022-06-01

**Authors:** Leen Oyoun Alsoud, Nelson C. Soares, Hamza M. Al-Hroub, Muath Mousa, Violet Kasabri, Nailya Bulatova, Maysa Suyagh, Karem H. Alzoubi, Waseem El-Huneidi, Bashaer Abu-Irmaileh, Yasser Bustanji, Mohammad H. Semreen

**Affiliations:** 1College of Pharmacy, University of Sharjah, Sharjah P.O. Box 27272, United Arab Emirates; u20105723@sharjah.ac.ae (L.O.A.); nsoares@sharjah.ac.ae (N.C.S.); kelzubi@sharjah.ac.ae (K.H.A.); 2Sharjah Institute for Medical Research, University of Sharjah, Sharjah P.O. Box 27272, United Arab Emirates; halhroub@sharjah.ac.ae (H.M.A.-H.); welhuneidi@sharjah.ac.ae (W.E.-H.); 3Research Institute of Science and Engineering, University of Sharjah, Sharjah P.O. Box 27272, United Arab Emirates; mmousa2@sharjah.ac.ae; 4School of Pharmacy, The University of Jordan, Amman 11942, Jordan; v.kasabri@ju.edu.jo (V.K.); nyounes@ju.edu.jo (N.B.); m.suyagh@ju.edu.jo (M.S.); 5College of Medicine, University of Sharjah, Sharjah P.O. Box 27272, United Arab Emirates; 6Hamdi Mango Center for Scientific Research, The University of Jordan, Amman 11942, Jordan; b.aburmela@ju.edu.jo

**Keywords:** metabolic syndrome, untargeted metabolomics, UHPLC-ESI-QTOF-MS, metabolites, metabolic pathways, MetaboAnalyst, metabolic profiling

## Abstract

Metabolic syndrome (MetS) is a disorder characterized by a group of factors that can increase the risk of chronic diseases, including cardiovascular diseases and type 2 diabetes mellitus (T2D). Metabolomics has provided new insight into disease diagnosis and biomarker identification. This cross-sectional investigation used an untargeted metabolomics-based technique to uncover metabolomic alterations and their relationship to pathways in normoglycemic and prediabetic MetS participants to improve disease diagnosis. Plasma samples were collected from drug-naive prediabetic MetS patients (*n* = 26), normoglycemic MetS patients (*n* = 30), and healthy (normoglycemic lean) subjects (*n* = 30) who met the inclusion criteria for the study. The plasma samples were analyzed using highly sensitive ultra-high-performance liquid chromatography electrospray ionization quadrupole time-of-flight mass spectrometry (UHPLC-ESI-QTOF-MS). One-way ANOVA analysis revealed that 59 metabolites differed significantly among the three groups (*p* < 0.05). Glutamine, 5-hydroxy-L-tryptophan, L-sorbose, and hippurate were highly associated with MetS. However, 9-methyluric acid, sphinganine, and threonic acid were highly associated with prediabetes/MetS. Metabolic pathway analysis showed that arginine biosynthesis and glutathione metabolism were associated with MetS/prediabetes, while phenylalanine, D-glutamine and D-glutamate, and lysine degradation were highly impacted in MetS. The current study sheds light on the potential diagnostic value of some metabolites in metabolic syndrome and the role of their alteration on some of the metabolic pathways. More studies are needed in larger cohorts in order to verify the implication of the above metabolites on MetS and their diagnostic value.

## 1. Introduction

Metabolic syndrome (MetS) is a disorder characterized by a group of factors that can increase the risk of chronic diseases, including cardiovascular (CV) disease and type 2 diabetes mellitus (T2D). MetS factors include abdominal obesity, dyslipidemia, increased blood pressure, and hyperglycemia [1]. MetS can be described as syndrome X, insulin resistance (IR) syndrome, or obesity dyslipidemia syndrome. The prevalence of MetS is positively associated with age and abdominal obesity [2]. Although excess visceral fat is concerning, abdominal subcutaneous adipose tissue and total body fat can also contribute to MetS complications, according to the national institute of health (NIH). Abdominal obesity is defined as a waist circumference of more than 88 cm in women and 102 cm in men. Furthermore, obese patients with a genetic predisposition, a sedentary lifestyle, or a disproportionate body fat distribution are more prone to developing T2D or CV diseases. In addition, these conditions are correlated with IR, which affects glucose and fatty acid metabolism [3].

There are several definitions for MetS, including those of the National Cholesterol Education Program Adult Treatment Panel III (NCEP ATP III), World Health Organization (WHO), International Diabetes Federation (IDF), European Group for the Study of IR (EGIR), and American Association of Clinical Endocrinologists (AACE) [4]. However, based on IDF, which we applied in this study, the specific diagnosis of MetS includes at least two of the following: fasting glucose ≥ 100 mg/dL (or receiving drug therapy for hyperglycemia), high-density lipoprotein (HDL) cholesterol <40 and <50 mg/dL for men and women, respectively (or receiving drug therapy for reduced HDL-C), triglycerides ≥ 150 mg/dL (or receiving drug therapy for hypertriglyceridemia), waist circumference ≥ 88 cm or 102 cm for females and males, respectively, blood pressure ≥ 130/85 mmHg (or receiving drug therapy for hypertension [5].

T2D is a multifactorial, multicomponent metabolic disease. It is characterized primarily by the progressive loss of pancreatic β-cells and insulin resistance [6]. Other diabetes types include T1D (autoimmune β-cell destruction), gestational diabetes mellitus, monogenic diabetes syndromes, or secondary diabetes associated with other diseases or due to drug- or chemical induction (such as with glucocorticoid use) [7]. Prediabetes describes the phase when blood sugar levels are higher than normal but not yet high enough to be diagnosed as diabetes. Based on the American Diabetes Association (ADA) guidelines, the patient is considered prediabetic if the glycosylated hemoglobin (HbA1C) level is 5.7–6.4% or fasting blood glucose (FBG) is 100–125 mg/dL. However, if the HbA1C level is ≥6.5% or FBG is ≥126 mg/dL, the patient is considered diabetic [8].

In recent years, omics technologies (metabolomics, proteomics, transcriptomics, and genomics) have shown quick advancement as a method of choice for early disease diagnosis [9]. Metabolites are small molecules that represent an organism or a cell metabolome and are linked to the functions of the cell via biomarkers [10]. Nowadays, advances in biomarker discovery are based on the highly developed technology related to liquid chromatography coupled with mass spectrometry (LC-MS/MS) used to detect and characterize biomolecules in complex matrices at a high level of sensitivity and selectivity. Furthermore, ultra-high-performance liquid chromatography electrospray ionization quadrupole time-of-flight mass spectrometry (UHPLC-ESI-QTOF-MS) is a highly sensitive, accurate, and robust analytical technology platform with high profiling identifications of small molecules in complexes [11]; thus, it was utilized in the current study for the biomarker analysis.

Several targeted and untargeted studies have identified particular metabolites for MetS and T2D. For instance, Zhong et al. observed several metabolite degradations associated with MetS [12]. In addition, an untargeted analysis of samples obtained from 228 participants with MetS showed significant alterations in metabolites [13]. On the other hand, during an eight-year follow-up period of a 1939 nondiabetic Korean cohort, 282 cases of incident T2D were identified, in which 22 metabolites were significantly associated with T2D risk [14].

This study aims to identify prediabetic biomarkers among MetS patients utilizing UHPLC-ESI-QTOF-MS in a Jordanian cohort. Our study sheds light on the biomarkers of the early diagnosis of prediabetes associated with MetS and their relation to metabolic pathways to enhance our understanding of the difference between normoglycemic MetS and prediabetic MetS.

## 2. Results

### 2.1. Participant and Blood Sample Characteristics

All study participants were Jordanians, with 46 females (53.5%) and 40 males (46.5%). The average age was 48.75 ± 12.87, with a statistically significant difference between the MetS groups and the control group (*p* < 0.001) (Table 1A). The prediabetic MetS group presented significantly higher values of HbA1c (*p*2 and *p*3 < 0.001) than both normoglycemic MetS and control groups. The two MetS groups showed significantly higher values of diastolic blood pressure (DBP) (*p*1 = 0.002, *p*2 < 0.001), systolic blood pressure (SBP) (*p*1 and *p*2 < 0.001), triglycerides (TG) (*p*1 and *p*2 < 0.001), and lower values of HDL-C (*p*1 and *p*2 < 0.001) than the control group did. The prediabetic MetS group showed significantly higher values of DBP (*p*3 = 0.019) and TG (*p*3 = 0.008) than the normoglycemic MetS group did. The prediabetic MetS group had significantly higher low-density lipoprotein cholesterol LDL-C (*p*2 = 0.028) than the control group did. In both MetS groups, waist circumference (WC) and body mass index (BMI) were significantly higher than in the control group. Both MetS groups had significantly higher values of MetS-surrogate IR (MetS-IR), triglyceride glucose (TyG) Index, TyG-BMI, and TyG-WC (*p*1 and *p*2 < 0.001) than the control group did (Table 1B).

### 2.2. Metabolic Changes

A total of 99 metabolites were found, among which 59 were statistically significant (*p* < 0.05) in the three examined groups according to the ANOVA test. The sparse partial least squares-discriminant analysis (sPLS-DA) showed minimal overlapping, indicating a difference between the three groups (Figure 1).

MetaboAnalyst 5.0 software was used to calculate the fold change between every two groups. The direction of comparison was as follows: normoglycemic MetS/control; prediabetic MetS/control; prediabetic MetS/normoglycemic MetS. Among the 99 metabolites detected, 35 differed significantly when comparing the control group with normoglycemic MetS (22 upregulated vs. 13 downregulated) (Table 2). Specifically, the four most upregulated metabolites in the normoglycemic MetS group were hippurate (hippuric acid), L-sorbose, homoveratric acid, and 5-hydroxy-L-tryptophan (Figure 2). In contrast, 3-methylxanthine, 3,4,5 trimethoxy cinnamic acid, 9-methyluric acid, and acetic acid were highly downregulated in the normoglycemic MetS group compared to those in the control group (Figure 2).

*T*-test analysis identified 57 significantly different metabolites between the control and prediabetic MetS groups (39 upregulated vs. 18 downregulated) (Table 3). Specifically, hippurate, L-sorbose, 5-hydroxy L-tryptophan, and homoveratric acid were highly upregulated in the prediabetic MetS group compared to those in the control group (Figure 3). In contrast, 3-methylxanthine, 9-methyluric, 3,4,5- trimethoxycinnamic acid, and 3-indolepropionic acid were highly downregulated in the prediabetic MetS group compared to those in the control group (Figure 3).

The *t*-test identified 18 metabolites as significantly different between the normoglycemic MetS and prediabetic MetS groups (upregulated 8 vs. 10 downregulated) (Table 4). Specifically, threonic acid, indolelactic acid, trimethylamine, and 5-hydroxy L-tryptophan were highly upregulated in the prediabetic MetS group compared to those in the normoglycemic MetS group (Figure 4). In contrast, 9-methyluric acid, sphinganine, vitamin D3, and deoxycholic acid glycine conjugate were highly downregulated in the prediabetic MetS group compared to those in the normoglycemic MetS, as indicated in the volcano plot (Figure 4).

### 2.3. Functional Analysis Pathway Changes

The pathway analysis’ modules on the MetaboAnalyst 5.0 (http://www.metaboanalyst.ca/, accessed on 20 April 2022) were performed by importing the set of significantly dysregulated metabolites matching the human metabolome database (HMDB), PubChem, and KEGG database for every two groups and then categorizing the metabolites based on the KEGG database. The first pathway analysis module was created for the significantly dysregulated metabolites among the control group and the normoglycemic MetS. Arginine biosynthesis (*p*-value = 0.003), D-glutamine and D-glutamate metabolism (*p*-value = 0.003), phenylalanine metabolism (*p*-value = 0.026), aminoacyl-tRNA biosynthesis (*p*-value = 0.032), and glyoxylate and dicarboxylate metabolism (*p*-value = 0.01) were significantly identified as the metabolic pathways in which altered metabolites between the groups were implicated in the normoglycemic MetS group compared to those in the control group (Figure 5A). The second module for the pathway analysis was constructed on the significantly dysregulated metabolites of the control group with the prediabetic MetS group. The latter group showed that arginine biosynthesis (*p*-value = 0.005), the D-glutamine and D-glutamate metabolism (*p*-value = 0.009), phenylalanine metabolism (*p*-value = 0.026), aminoacyl-tRNA biosynthesis (*p*-value = 0.007), glyoxylate and dicarboxylate metabolism (*p*-value = 0.05), nitrogen metabolism (*p*-value = 0.009), and lysine degradation (*p*-value = 0.026) were significantly identified as the metabolic pathways in which altered metabolites between the groups were implicated in the prediabetic MetS group compared to those in the control group (Figure 5B). Finally, the third module was designed for the normoglycemic MetS group and the prediabetic MetS group. Arginine biosynthesis (*p*-value = 0.006) and glutathione metabolism (*p*-value = 0.025) were significantly altered in the prediabetic MetS group compared to those in the normoglycemic MetS group (Figure 5C). A random sample from each of the three groups of the UHPLC-QTOF-MS base peaks chromatograms is shown in (Figure 6A–C).

## 3. Discussion

We performed a cross-sectional study that compared MetS patients with prediabetes vs. normoglycemic MetS patients and controls using untargeted LC-MS/MS. A total of 57 metabolites were significantly associated with prediabetes in MetS patients in relation to the control group. MetS is insulin-resistance-related, which, in turn, is associated with renal diseases. Therefore, the metabolites considered as biomarkers of renal illnesses can be utilized to diagnose MetS [15]. Notably, hippurate has been reported as a uremic toxin associated with disease progression in patients with chronic kidney disease [16]. Our results showed that hippurate was highly upregulated in normoglycemic MetS (144.6-fold) and prediabetic MetS patients (169-fold) compared to that in the control, emphasizing the likelihood of its association with MetS. Furthermore, this finding agrees with another metabolomic study that also revealed that hippurate is a potential biomarker in prediabetic and diabetic models [17].

Carbohydrates, including fructose and glucose, are associated with MetS [18]. A meta-analysis of 18 studies found an increased risk of MetS with carbohydrate consumption [19]. Fructose is retained within the small intestine and metabolized within the liver, where it invigorates fructolysis, glycolysis, lipogenesis, and glucose generation, leading to MetS [20]. Interestingly, in contrast to the previous findings, we found that L-sorbose was the one to be highly associated with MetS when comparing the normoglycemic MetS group (80.8-fold) and the prediabetic MetS group (112.3-fold) vs. the control group. The mass spectra and chemical structure of L-sorbose are shown in (Appendix A).

The metabolite 9-methyluric acid was highly downregulated in the prediabetic MetS group vs. the control group (0.06-fold) and vs. the normoglycemic MetS group (0.26-fold). As 9-methyluric acid is a methyl derivative of the antioxidant (uric acid), its low level can result in oxidative stress and endothelial dysfunction, which leads to diabetes [21,22]. Our results indicate the likelihood of 9-methyluric acid being a potential biomarker for T2D, specifically in MetS/prediabetic patients, especially since 7-methyluric acid (which is another methyl derivative of uric acid) has been reported as a potential biomarker in type 2 diabetes [23].

Threonic acid was upregulated in the prediabetic MetS group compared to that in the normoglycemic MetS group (1.96-fold), indicating threonic acid as a possible biomarker for prediabetes. Threonic acid is the main breakdown product of ascorbic acid, and it has been reported before in a metabolomic profile study to be a potential biomarker in diabetic retinopathy [24], suggesting that threonic acid could be a potential biomarker for T2D, specifically in MetS/prediabetes patients as per our data.

A previous study observed higher levels of 5-hydroxy L-tryptophan, 5-hydroxyindoleacetic acid, kynurenic acid, 3-hydroxykynurenine, and 3-hydroxyanthranilic acid in T2D patients [25]. This is in agreement with our results which showed an increase in 5-hydroxy L-tryptophan among the prediabetic MetS group compared to that in the control (7.8-fold) and compared to that in the normoglycemic MetS group (1.5-fold). The mass spectra and chemical structure of 5-hydroxy L-tryptophan are shown in (Appendix A).

Sphinganine was downregulated in the prediabetic MetS group compared to the control group (0.3-fold) and compared to the normoglycemic MetS group (0.27-fold). Sphinganine inhibits LDL-induced cholesterol esterification, causing unesterified cholesterol to concentrate in perinuclear vesicles and blocking post-lysosomal cholesterol transport. Endogenous sphinganine has been postulated as a possible cholesterol transport inhibitor in Niemann–Pick Type C (NPC) illness [26,27]. High cholesterol-to-HDL-cholesterol (HDL-C) ratio can indicate T2D, as elevated cholesterol levels impair glucose tolerance [28]. The downregulation of sphinganine was in agreement with a previous metabolomic study that reported that sphinganine was downregulated in diabetic patients compared to that in normal patients [29]. Taken together, sphinganine could be a potential biomarker for prediabetic patients, particularly those with MetS. The utilized instrument can analyze a wide range of metabolites with high resolution and accuracy. Our research utilized UHPLC-ESI-QTOF-MS to identify plasma biomarkers that characterize the association between MetS and prediabetes. The untargeted approach was highly effective in identifying biomarkers of phenotype studies. However, the relatively small number of patients may be considered a limitation in probing associations and the age and the BMI, which were not matched among the groups. A future targeted study related to our findings can be done to validate the obtained results.

### 3.1. The Arginine Biosynthesis Pathway

The arginine pathway was significantly perturbed when comparing each normoglycemic MetS group and prediabetic MetS group with the control (Figure 5A–C). Additionally, it was impacted in the prediabetic MetS group vs. the normoglycemic MetS group. Arginine plays a vital role in several biological processes, and it has been related to diabetes pathogenesis, where arginine biosynthesis has been found to affect glucose homeostasis, lipolysis, hormone levels, insulin resistance, and fetal programming in the early stages. The L-arginine–nitric oxide pathway, which can activate cell signal proteins, is a plausible signaling mechanism for the positive effects of L-arginine [30]. In humans, arginine plays an indispensable role as a necessary substrate in the urea cycle, which involves using arginine to transport nitrogenous wastes. However, the decrease in arginine availability for the liver slows down ureagenesis [31], which can explain the downregulation of urea in our results. Furthermore, the enrichment analysis performed between the normoglycemic MetS vs. control, prediabetic MetS vs. normoglycemic MetS, and prediabetic MetS vs. control groups showed that the urea metabolite was significantly impacted (Appendix A).

### 3.2. The D-Glutamine and D-Glutamate Metabolism Pathway

D-glutamine and D-glutamate metabolism was significantly impacted when comparing each prediabetic MetS and normoglycemic MetS group with the control group (Figure 5A,B). Previous studies have shown glutamine and aromatic amino acids to be higher in T2D patients [32]. Our results validate these findings, with glutamine (4.8-fold), tryptophan (1.1-fold), and tyrosine (1.3-fold) levels being significantly higher in the prediabetic MetS group than in the control group (*p* < 0.05). In addition, glutamine was shown to be upregulated in the normoglycemic MetS group relative to that in the control group (3.5-fold), emphasizing its association with MetS (Table 2). Li Y et al. reported several pathways to be associated with MetS; however, in accordance with our results, D-glutamine and D-glutamate metabolism was impacted [33]. In our results, D-glutamine and D-glutamate metabolism was highly affected by L-glutamine and L-glutamate (L-glutamic acid) metabolites, which tended to be upregulated as reported above. Additionally, the hyperinsulinism/hyperammonemia syndrome could develop due to increased glutamate dehydrogenase activity [34]. The evidence mentioned above emphasizes the likelihood of the association between MetS and D-glutamine and D-glutamate metabolism.

### 3.3. The Phenylalanine Metabolism Pathway

Phenylalanine metabolism was impacted when comparing the normoglycemic MetS group with the control group (Figure 5A). Li Y et al. also observed phenylalanine metabolism to be associated with MetS [33]. Our data showed that phenylalanine metabolism was altered due to phenylacetaldehyde and hippurate both being upregulated in the normoglycemic MetS group (1.26-fold and 144.6-fold, respectively). Additionally, Adams and fellows reported phenylalanine and tyrosine metabolism alteration in obese, insulin-resistant, and T2DM patients [35]. A previous study found cysteine, phenylalanine, and tyrosine to have a significantly higher level of nitrotyrosine, which was used to assess oxidative stress among T2DM patients [36]. In addition, aromatic amino acids (phenylalanine and tyrosine) have been associated in other studies with insulin resistance and an increased risk of developing diabetes [32,37]. Interestingly, we observed a phenylalanine metabolism alteration to be associated with MetS. Phenylalanine hydroxylase (PAH) is an enzyme expressed in the liver and kidney responsible for phenylalanine metabolism [38]. The occurrence of such metabolic disease help in explaining the pathway’s impact to MetS, as noted in our data.

### 3.4. The Lysine Degradation Pathway

Lysine degradation was altered when comparing the prediabetic MetS group with the control group (Figure 5B). In T2DM patients, lysine treatment was found to have a therapeutic effect in reducing the formation of glycated lysozyme [39,40]. Protein glycation is a well-known process linked to long-term hyperglycemia, which has been linked to various pathophysiological diseases, including cancer, inflammation, metabolic dysfunctions, and aging [41]. According to our data, the pathway was impacted mainly by downregulated (0.66-fold and 0.85-fold, respectively) saccharporine and L-hydroxylysine, and L-pipecolic acid, which was upregulated in the prediabetic MetS group (1.2-fold).

### 3.5. The Glutathione Metabolism Pathway

Glutathione metabolism was perturbed when comparing the prediabetic MetS group with the normoglycemic MetS group (Figure 5C; Appendix A). Glutathione (γ-glutamyl-cysteinyl glycine) is a major intracellular antioxidant that reduces oxidative stress. There has been evidence that erythrocyte glutathione (GSH) concentrations are decreased in patients with T2D [42]. A study on patients with T2D showed similar concentrations of total erythrocyte glutathione levels compared to those in nondiabetic controls, but reduced GSH. Higher levels of oxidized glutathione (GSSG) indicate increased uptake [43], by which impaired glutathione metabolism can lead to hyperglycemia. Oxidative stress is significantly increased in microvascular complications, but it is unclear whether patients with complications have more glutathione metabolic disorders than patients with uncomplicated diabetes do [42]. In our study, glutathione metabolism was mainly impacted by L-glutamate and pyroglutamic acid, both upregulated (1.14-fold and 1.3-fold, respectively) in the prediabetic MetS group.

### 3.6. Aminoacyl tRNA Biosynthesis

Aminoacyl tRNA biosynthesis was perturbed when comparing the two MetS groups to the control group. As a part of protein synthesis, aminoacyl-tRNA synthetases pair tRNAs with amino acids for mRNAs decoding per the genetic code [44]. Synthetases also appear to be important in many other cellular processes, with profound implications in health and diseases, including metabolic and autoimmune disorders [45]. In concordance with our results, Zhang et al. observed aminoacyl-tRNA biosynthesis pathway perturbation among MetS patients [46]. Furthermore, and in agreement with our findings, aminoacyl-tRNA biosynthesis was also found to be altered in a metabolomic profiling analysis in prediabetics [47]. Taken together, our results and previous findings emphasize the association of aminoacyl tRNA with MetS.

### 3.7. Glyoxylate and Dicarboxylate Metabolism

In the current study, glyoxylate and dicarboxylate metabolism was also altered in the two MetS groups compared to that in the control group. This finding agrees with a recently published study using metabolic profiling analysis in prediabetes, where they observed glyoxylate and dicarboxylate metabolism to be among the significantly perturbed metabolic pathways [47]. A review of cardiometabolic diseases, including MetS, reported several pathways to be altered, of which glyoxylate and dicarboxylate metabolism and aminoacyl-tRNA biosynthesis were highly significant [48]. Further, altered metabolism of glyoxylate and dicarboxylate has been linked to mitochondrial dysfunction, which negatively affects the ability to detoxify reactive oxygen species (ROS) in elderly females [49]. The increase in ROS causes cellular damage and further leads to oxidative stress [50], which is often associated with MetS [51]. In our results, glyoxylate and dicarboxylate metabolism were highly affected by L-glutamine, L-glutamate, and threonine metabolites, which tended to be upregulated as mentioned above and thus may be linked to MetS occurrence.

### 3.8. Nitrogen Metabolism

Our results showed that nitrogen metabolism was also impacted in the two MetS groups compared to that in the control group. This finding also agrees with a recent study done by Li et al., as they found that nitrogen metabolism was altered in the simple diabetic group compared to that in those with impaired fasting glucose, which represent a prediabetic group [52]. This also agrees with another study reporting that the nitrogen metabolism pathway and its components are potential effectors of the earliest stages of type 2 diabetes pathophysiology [53].

## 4. Materials and Methods

### 4.1. Population and Study Design

The patients’ samples were collected from Jordan University Hospital, and the metabolomics study was conducted at the Sharjah Institute of Medical Research (SIMR). The samples were divided into three groups: control group (30), normoglycemic MetS patients (30), and prediabetic MetS patients (26).

The inclusion criteria for all the groups were age between 18 and 75 years old and agreement to participate in the study. The inclusion criteria of the control group were normal weight (body mass index (BMI) < 25 kg/m^2^) and normal plasma glucose (HbA1C < 5.7%). For the normoglycemic MetS group, the inclusion criteria were normal plasma glucose (HbA1C < 5.7%, FPG < 100 mg/dL) with ≥2 MetS components according to the IDF [4] and being overweight (BMI > 25 kg/m^2^) or obese (BMI > 30 kg/m^2^). For the prediabetic MetS group, the inclusion criteria were ≥2 MetS components according to the IDF [4] with newly diagnosed prediabetes according to the ADA guidelines [8] and being anti-hyperglycemic drug-naive and either overweight or obese. All included patients fasted for 10–12 h before sample collection. In addition, the lipid profile, FBG, and HbA1C were tested.

The exclusion criteria were: (1) nonfasting candidates; (2) pregnant or breastfeeding females; (3) participants who received any prior anti-diabetic agent either for diabetes itself or for any other condition associated with hyperglycemia (sulfonylureas, meglitinides, biguanides, thiazolidinediones, alpha-glucosidase inhibitors), antihyperlipidemic agent (hydroxymethylglutaryl-CoA reductase inhibitors, fibric acids derivatives, bile acid-binding resins, nicotinamides, cholesterol absorption inhibitors), or antihypertensives including diuretics, calcium channel blockers (CCB), angiotensin-converting enzyme (ACE) inhibitors, adrenergic receptor antagonists, renin inhibitors, vasodilators, or angiotensin II receptor blockers (ARBs); (4) clinical evidence of autoimmune or life-threatening disease (alcohol/drug abuse/recently diagnosed untreated endocrine disorder); (5) individuals with known autoimmune diseases such as inflammatory bowel disease or obesity secondary to endocrine derangement other than DM.

The selected participants were randomly approached in the patients’ waiting area or the triage station in outpatient family medicine clinics. Those who were evident to meet the inclusion criteria were asked to participate in the study, and upon their approval, informed consent was given. The study commenced after approval from the Research Ethics Committee of Jordan University Hospital. The study was conducted according to the guidelines of the Declaration of Helsinki, and the volunteers have been informed in detail about the study before signing the consent forms.

### 4.2. Collection of Samples

A total of 86 individuals were recruited in the study. Plasma samples were collected from 30 healthy patients, 30 normoglycemic MetS patients, and 26 prediabetic MetS patients. Plasma was obtained after collecting samples into heparinized tubes followed by centrifugation for 5 min (14,000 rpm). Plasma samples were subsequently stored at −80 °C and shipped to the Research Institute University of Sharjah for further analysis.

### 4.3. Preparation of the Samples for Metabolomics Extraction

Samples were divided into Eppendorf of 100 µL each, 300 µL of methanol (Wunstorfer Strasse, Seelze, Germany) was added, followed by vortex and incubation at –20 °C for 2 h. Next, the samples were vortexed and then centrifuged at 14,000 rpm for 15 min. Then, the supernatant was evaporated using Speed vacuum evaporation at 35–40 °C. Next, a quality control (QC) sample was prepared by pooling the same volume of each sample to evaluate the reproducibility of the analysis. The extract samples were then resuspended with 250 µL of 0.1% formic acid in Deionized Water-LC-MS CHROMASOLV from Honeywell (Wunstorfer Strasse, Seelze, Germany). Then, the supernatant was filtered using a 0.45 µm pore size hydrophilic nylon syringe filter for LC-MS/MS analysis using 100 µL of the prepared sample collected in an insert within LC glass vials.

### 4.4. Ultra-High-Performance Liquid Chromatography Coupled to Electrospray Ionization and Quadrupole Time-of-Flight Mass Spectrometry (UHPLC-ESI-QTOF-MS)

An ultra-high-performance liquid chromatography system (UHPLC) (Bruker Daltonik GmbH, Bremen, Germany) coupled to a quadrupole time-of-flight mass spectrometer (QTOF) was utilized to perform the LC-MS/MS analysis. The system was equipped with an electrospray ionization (ESI) source, a solvent delivery systems pump (HPG 1300), an autosampler, and a thermostat column compartment. Windows 10 Enterprise 2016 LTSB was used as the computer operating system. The data management software was Bruker Compass HyStar 5.0 SR1 Patch1 (5.0.37.1), Compass 4.1 for otofSeries, otof Control Version 6.2.

Mobile phases A (water with 0.1% formic acid) and B (acetonitrile with 0.1% formic acid) were employed. The gradient program was: 0–2 min, 99% A: 1% B; 2–17 min, 99–1% A: 1–99% B; 17–20 min, 99% B: 1% A. The flow rate was fixed at 0.25 mL/min. Subsequently, 20–20.1 min 99% B to 99% A; 20.1–28.5 min, 99% A: 1% B at 0.35 mL/min flow rate; 28.5–30 min; 99% A: 1% B at 0.25 mL/min. A 10 μL aliquot of the sample was injected, and the separation was performed on a Hamilton^®^ Intensity Solo 2 C18 column (100 mm × 2.1 mm × 1.8 μm) at a column oven temperature set at 35 °C.

The ESI source conditions for every injection were as follows: the drying gas flow rate was 10.0 L/min at a temp of 220 °C; the capillary voltage was set at 4500 V; a nebulizer pressure of 2.2 bar. For MS2 acquisition, the collision energy stepping fluctuated between 100 and 250% set at 20 eV and an End Plate offset of 500 V [54]. Sodium formate was used as a calibrant for the external calibration step.

The acquisition involved two segments; auto MS scan, which ranged from 0 to 0.3 min for the calibrant sodium formate, and auto MS/MS, which included fragmentation and ranged from 0.3 to 30 min. The acquisition in both segments was performed using the positive mode at 12 Hz. The automatic in-run mass scan range was from 20 to 1300 *m*/*z*, the width of the precursor ion was ±0.5, the number of precursors was 3, the cycle time was 0.5 s, and the threshold was 400 cts. Active exclusion was excluded after 3 spectra and released after 0.2 min.

### 4.5. Data Processing and Analysis

The data was processed using MetaboScape^®^ 4.0 software (Bruker Daltonics, Billerica, MA, USA) [55]. Bucketing parameters of the processed data in T-ReX 2D/3D workflow were as follows: intensity threshold of 1000; peak length of 7 spectra; utilizing peak area for quantifying the feature. The calibration for mass spectra was done in 0–0.3 min with features ranging in at least 30 to 172 samples. However, the auto MS/MS scan was done using the average method. The scan parameters were at a retention time range of 0.3 to 25 min and a mass range of 50 to 1000 *m*/*z*. Each sample was analyzed in duplicate by LC-QTOF, which generated a data set of 15,000 features across 86 samples of the three examined groups. Identification of metabolites was based on mapping the MS/MS spectra and retention time in the HMBD 4.0, an annotated resource designed to satisfy the needs of the metabolomics community [56]. The compounds with MS/MS were identified using library matching through the annotation process. Then, the selected metabolites were filtered by choosing the set with a higher annotation quality score (AQ score) representing the best retention time values, MS/MS score, *m*/*z* values, mSigma, and analyte list spectral library. A total of 99 distinct metabolites were selected after filtration (Appendix A). The chemical structures of the distinct metabolites are shown in (Appendix A). The quantification of the data matrix was based on the peak intensity of each metabolite. The metabolite datasets included only the significant compounds registered in the HMDB 4.0 with *p* < 0.05

The metabolite datasets were exported as a CSV file and imported into MetaboAnalyst 5.0 software (Mcgill University, Montreal, QC, Canada), a comprehensive platform for metabolomics data analysis [57]. The sPLS-DA was performed using MetaboAnalyst to select the most discriminative features in the examined group to aid in classifying the samples [56]. The false discovery rate (FDR) method was utilized to correct multiple hypothesis testing and reduce the rate of false positives.

### 4.6. Metabolic Pathway and Statistical Analysis

The enrichment metabolite sets and pathway analysis were processed using MetaboAnalyst 5.0 [57]. For statistical analysis, univariate statistical tests: analysis of variance (ANOVA) for all three groups (control, normoglycemic MetS, and prediabetic MetS), and unpaired *t*-tests for every two groups (control and normoglycemic MetS; control and prediabetic MetS; normoglycemic MetS and prediabetic MetS) were employed using MetaboAnalyst 5.0. A *p*-value of <0.05 was considered statistically significant.

## 5. Conclusions

Our study demonstrated the metabolites and metabolic pathways in MetS patients with and without prediabetes among a Jordanian cohort. Glutamine, 5-hydroxy-L-tryptophan, L-sorbose, and hippurate were highly altered with MetS (both normoglycemic and prediabetic; *n* = 56). However, 9-methyluric acid, sphinganine, and threonic acid were highly associated with prediabetes, which might be particularly associated with MetS. In addition, arginine biosynthesis and glutathione metabolism were associated with MetS/prediabetes, while D-glutamine and D-glutamate, phenylalanine metabolism, aminoacyl tRNA biosynthesis, glyoxylate and dicarboxylate metabolism, and nitrogen metabolism were highly impacted in MetS. More studies are needed in larger cohorts in order to verify the implication of these metabolites on MetS and their diagnostic value.

## Figures and Tables

**Figure 1 metabolites-12-00508-f001:**
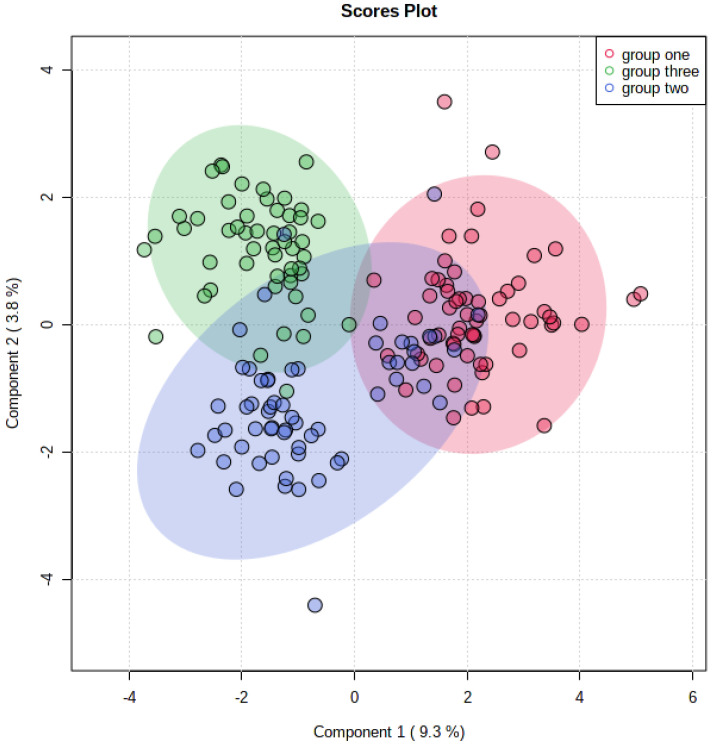
The sparse partial least squares-discriminant analysis (sPLS-Da) for all groups; group one: control; group two: normoglycemic MetS; group three: prediabetic MetS.

**Figure 2 metabolites-12-00508-f002:**
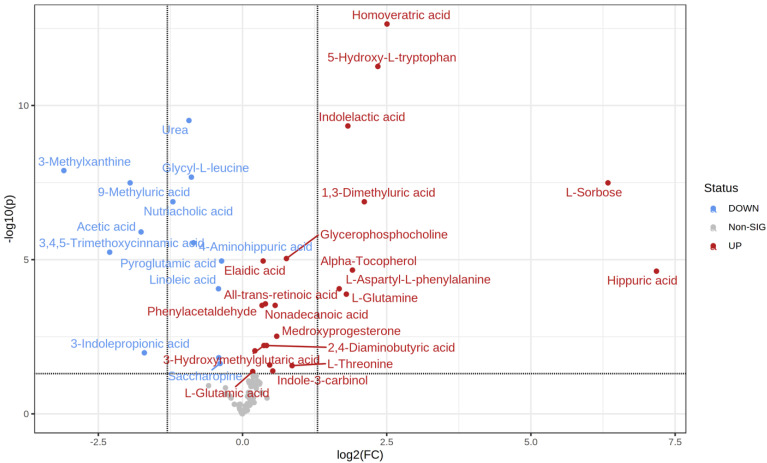
Volcano plot of the control group and the normoglycemic metabolic syndrome (MetS) group.

**Figure 3 metabolites-12-00508-f003:**
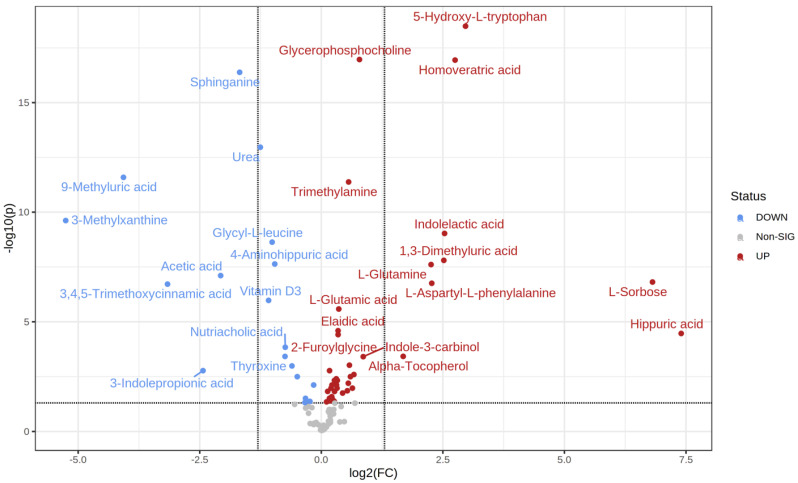
Volcano plot of the prediabetic MetS group and the controls.

**Figure 4 metabolites-12-00508-f004:**
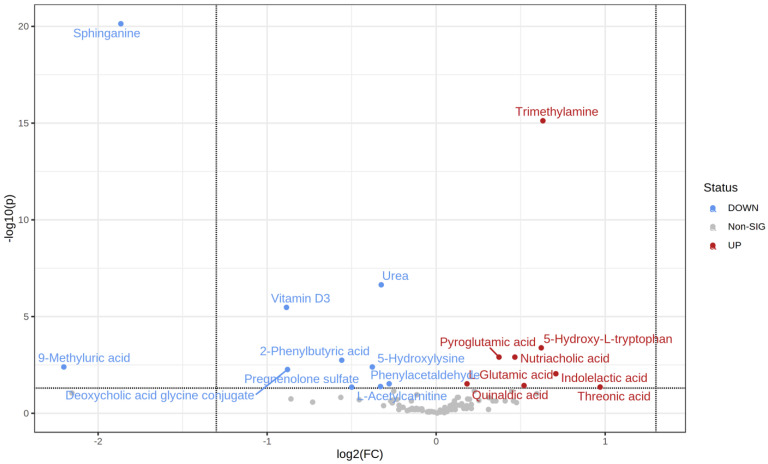
Volcano plot of the normoglycemic MetS and the prediabetic MetS groups.

**Figure 5 metabolites-12-00508-f005:**
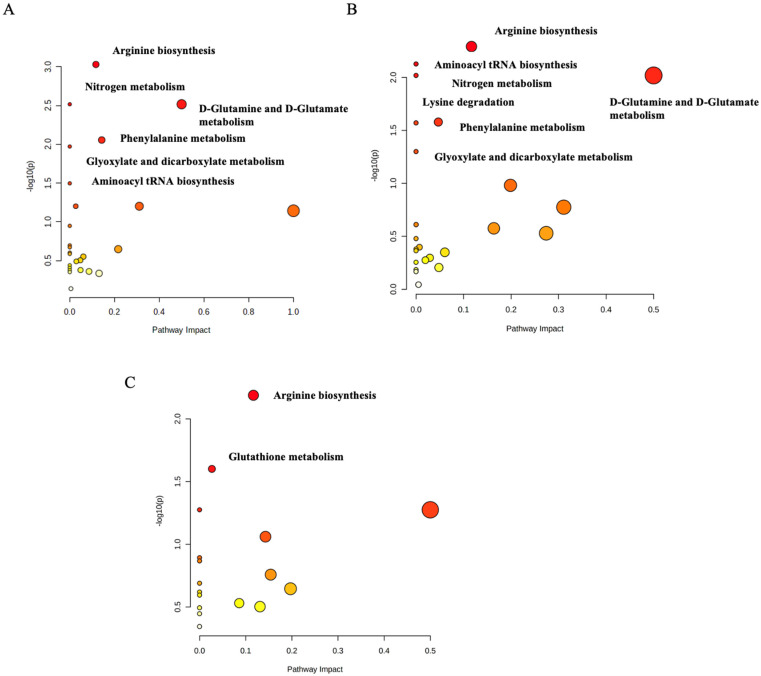
Pathway analysis of (**A**) the normoglycemic MetS group and the control group; (**B**) the controls and the prediabetic MetS group; (**C**) the normoglycemic MetS and the prediabetic MetS groups.

**Figure 6 metabolites-12-00508-f006:**
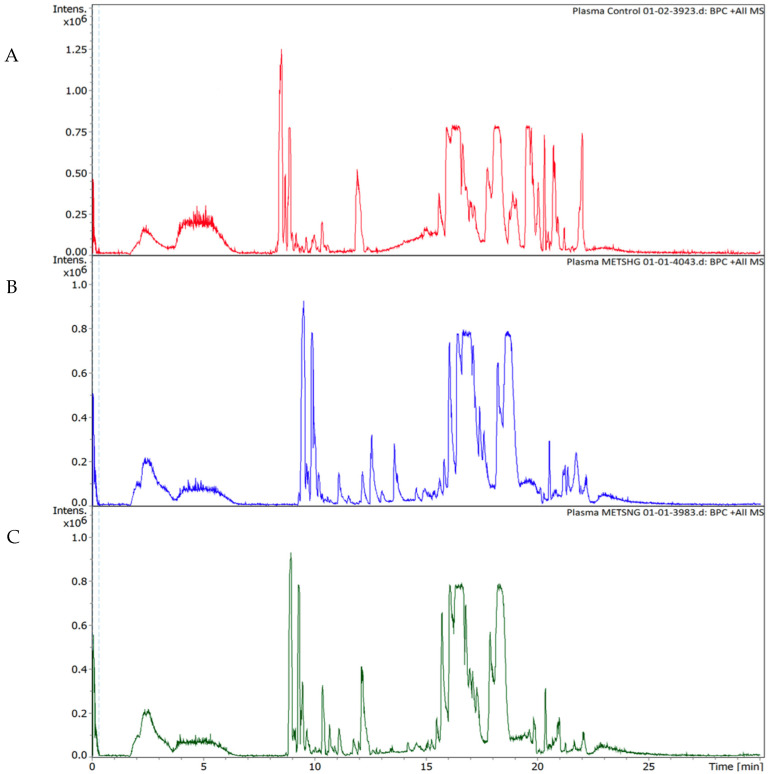
Positive ion base peak intensity chromatograms obtained from the analysis of (**A**) the control group, (**B**) the prediabetic MetS group, and (**C**) the normoglycemic MetS group.

**Table 1 metabolites-12-00508-t001:** Patients’ demographic (**A**) and clinical characteristics (**B**).

(**A**) **Patients’ Demographics**						
	**Total Sample** **Mean ± SD (*n* = 86)**	**Controls** **Group Mean ± SD (*n* = 30)**	**Normoglycemic (MetS)** **Mean ± SD (*n* = 30)**		**Prediabetic MetS** **Mean ± SD (*n* = 26)**	***p*-Value**
**Female, n (%)**	46 (53.5%)	16 (53.3%)	19 (63.3%)		11 (42.3%)	0.113
**Age (years)**	48.75 ± 12.87	37.83 ± 11.1	54.13 ± 10.72		53.73 ± 9.55	<0.001
(**B**) **Patients’ clinical characteristics**						
	**Control Group Mean ± SD (*n* = 30)**	**Normoglycemic MetS** **Mean ± SD (*n* = 30)**	**Prediabetic MetS** **Mean ± SD (*n* = 26)**	***p*1-Value**	***p*2-Value**	***p*3-Value**
**BMI (kg/m^2^)**	22.91 ± 2	32.74 ± 4.3	32.87 ± 3.8	<0.001	<0.001	1
**SBP** **(mmHg)**	111.23 ± 9.05	132.20 ± 11.61	138.17 ± 11.55	<0.001	<0.001	0.106
**DBP (mmHg)**	73.77 ± 6.82	81.43 ± 9.22	87.50 ± 8.93	0.002	<0.001	0.019
**HbA1C%**	5.13 ± 0.3	5.43 ± 0.243	6.65 ± 1.32	0.471	<0.001	<0.001
**FPG (mg/dL)**	88.18 ± 8.68	101.07 ± 15.95	124.31 ± 47.38	0.275	<0.001	0.009
**TG (mg/dL)**	76.33 ± 23.61	168.74 ± 47.14	221.84 ± 102.7	<0.001	<0.001	0.008
**LDL-C (mg/dL)**	117.1 ± 36.57	129.7 ± 36.57	141.03 ± 43.12	0.493	0.028	0.632
**HDL-C (mg/dL)**	58.4 ± 10.21	45.87 ± 8.68	45.40 ± 13.03	<0.001	<0.001	1
**LDL-C/HDL C ratio**	2.06 ± 0.53	2.86 ± 0.73	3.28 ± 1.15	<0.001	0.01	0.188
**WC (cm)**	77.37 ± 7.74	111.03 ± 10.5	115.2 ± 10.26	<0.001	<0.001	0.287
**Surrogate insulin resistance (sIR) indices**						
**MetS-IR**	13.08 ± 1.67	22.68 ± 3.018	24.55 ± 3.85	<0.001	<0.001	0.06
**TyG Index**	8.06 ± 0.39	9.0 ± 0.24	9.38 ± 0.62	<0.001	<0.001	0.005
**TyG-BMI**	185.14 ± 21.01	294.66 ± 37.93	308.38 ± 39.81	<0.001	<0.001	0.365
**TyG-WC**	625.3 ± 82.47	999.07 ± 87.59	1081.6 ± 127.91	<0.001	<0.001	0.007

MetS: metabolic syndrome; SD: standard deviation. Comparisons of means and *p*-values were obtained by the ANOVA test. Pairwise post hoc comparisons were done through Bonferroni. *p*1: MetS/normoglycemic group versus control; *p*2: MetS/preDM group versus control; *p*3: MetS/preDM group versus normoglycemic MetS group; BMI: body mass index; SBP: systolic blood pressure; DBP: diastolic blood pressure; HbA1C%: percent glycosylated hemoglobin; FPG: fasting plasma glucose; TG: triglyceride; LDL-C: low-density lipoprotein cholesterol; HDL-C: high-density lipoprotein cholesterol; LDL-C/HDL-C: low-density lipoprotein cholesterol to high-density lipoprotein cholesterol ratio; WC: waist circumference; TyG Index: triglyceride glucose (TG) index, TyG-BMI: TyG index-to-BMI ratio; TyG-WC: TyG index-to-WC ratio, MetS-IR: Metabolic Score for Insulin Resistance.

**Table 2 metabolites-12-00508-t002:** Statically significant metabolites among the normoglycemic MetS and the controls.

	*t*.Stat	*p*.Value	FDR	Fold Change
Hippuric acid	−5.0653	4.29 × 10^−6^	2.36 × 10^−5^	144.62
L-Sorbose	−7.0091	2.61 × 10^−9^	3.23 × 10^−8^	80.759
Homoveratric acid	−10.342	2.30 × 10^−15^	2.28 × 10^−13^	5.6759
5-Hydroxy-L-tryptophan	−9.366	1.10 × 10^−13^	5.42 × 10^−12^	5.0849
1,3-Dimethyluric acid	−6.4478	1.21 × 10^−8^	1.33 × 10^−7^	4.3219
Alpha-Tocopherol	−5.0201	3.74 × 10^−6^	2.18 × 10^−5^	3.748
Indolelactic acid	−8.028	1.86 × 10^−11^	4.61 × 10^−10^	3.547
L-Glutamine	−4.5216	2.79 × 10^−5^	0.00013146	3.4823
L-Aspartyl-L-phenylalanine	−4.5417	1.69 × 10^−5^	8.78 × 10^−5^	3.1962
L-Threonine	−2.6831	0.0091206	0.027362	1.8144
Glycerophosphocholine	−5.2978	1.30 × 10^−6^	9.19 × 10^−6^	1.6951
Medroxyprogesterone	−3.4628	0.00076961	0.0030477	1.5083
Nonadecanoic acid	−4.1575	7.15 × 10^−5^	0.00030394	1.4801
Indole-3-carbinol	−2.4999	0.013962	0.040654	1.4384
3-Hexenedioic acid	−2.681	0.0083979	0.025981	1.3861
2,4-Diaminobutyric acid	−3.2445	0.001658	0.0060793	1.3345
All-trans-retinoic acid	−4.2232	6.02 × 10^−5^	0.00027072	1.3167
3-Hydroxymethylglutaric acid	−3.2301	0.0016254	0.0060793	1.2938
Elaidic acid	−5.0285	1.79 × 10^−6^	1.10 × 10^−5^	1.2811
Phenylacetaldehyde	−4.1249	7.37 × 10^−5^	0.00030394	1.2643
2-Furoylglycine	−3.0855	0.002555	0.0090336	1.1606
L-Glutamic acid	−2.4676	0.015037	0.042534	1.1302
Pyroglutamic acid	5.0629	1.71 × 10^−6^	1.10 × 10^−5^	0.77785
Saccharopine	2.7334	0.0072891	0.023278	0.76389
Androsterone	2.9115	0.0045829	0.015124	0.75177
Linoleic acid	4.5263	1.77 × 10^−5^	8.78 × 10^−5^	0.749
4-Aminohippuric acid	5.4352	3.43 × 10^−7^	2.83 × 10^−6^	0.55491
Glycyl-L-leucine	6.6067	1.28 × 10^−9^	2.11 × 10^−8^	0.54098
Urea	8.2841	9.31 × 10^−12^	3.07 × 10^−10^	0.52527
Nutriacholic acid	6.4667	1.34 × 10^−8^	1.33 × 10^−7^	0.4331
3-Indolepropionic acid	3.0702	0.0030686	0.010476	0.30739
Acetic acid	5.8209	1.40 × 10^−7^	1.26 × 10^−6^	0.2955
9-Methyluric acid	6.6234	2.53 × 10^−9^	3.23 × 10^−8^	0.2591
3,4,5-Trimethoxycinnamic acid	5.4243	7.53 × 10^−7^	5.73 × 10^−6^	0.20262
3-Methylxanthine	7.1403	6.53 × 10^−10^	1.29 × 10^−8^	0.11684

FDR: false discovery rate.

**Table 3 metabolites-12-00508-t003:** Statically significant metabolites among the prediabetic MetS group and the controls.

	*t*.Stat	*p*.Value	FDR	Fold Change
Hippuric acid	−4.997	7.23 × 10^−6^	3.41 × 10^−5^	168.99
L-Sorbose	−6.5991	2.34 × 10^−8^	1.54 × 10^−7^	112.29
5-Hydroxy-L-tryptophan	−14.973	3.26 × 10^−21^	3.23 × 10^−19^	7.8145
Homoveratric acid	−13.381	3.46 × 10^−19^	1.14 × 10^−17^	6.7333
Indolelactic acid	−8.0659	8.53 × 10^−11^	9.39 × 10^−10^	5.7961
1,3-Dimethyluric acid	−7.1566	1.76 × 10^−9^	1.58 × 10^−8^	5.7356
L-Aspartyl-L-phenylalanine	−6.2668	2.84 × 10^−8^	1.76 × 10^−7^	4.8287
L-Glutamine	−7.0562	3.24 × 10^−9^	2.47 × 10^−8^	4.7812
Alpha-Tocopherol	−4.1864	9.14 × 10^−5^	0.00037703	3.2137
Indole-3-carbinol	−4.082	0.00010219	0.00038912	1.8178
Glycerophosphocholine	−11.298	2.17 × 10^−19^	1.07 × 10^−17^	1.7227
Nonadecanoic acid	−3.5226	0.00079411	0.002536	1.5918
3-Hexenedioic acid	−2.9064	0.0047055	0.010587	1.5583
Medroxyprogesterone	−3.3856	0.0010441	0.0031729	1.5147
Heptadecanoic acid	−3.8583	0.00025945	0.00095133	1.4928
Trimethylamine	−8.3473	2.96 × 10^−13^	4.19 × 10^−12^	1.4759
Quinaldic acid	−3.17	0.0024236	0.0063142	1.4694
3-Hydroxymethylglutaric acid	−2.8211	0.0063286	0.013923	1.4489
L-Norleucine	−2.6868	0.0085954	0.017728	1.355
L-Glutamic acid	−5.3425	5.02 × 10^−7^	2.62 × 10^−6^	1.2829
2-Furoylglycine	−4.6727	8.55 × 10^−6^	3.85 × 10^−5^	1.2707
Elaidic acid	−4.7976	5.15 × 10^−6^	2.55 × 10^−5^	1.2702
Benzoic acid	−3.2383	0.001675	0.0047047	1.2563
o-Tyrosine	−2.9137	0.0045066	0.010587	1.2511
m-Coumaric acid	−3.0439	0.0029627	0.0075208	1.2423
Pantothenic acid	−3.3059	0.0013618	0.0039652	1.238
2,4-Diaminobutyric acid	−3.1673	0.0020146	0.0053904	1.2297
L-Proline	−2.7595	0.0070984	0.014952	1.2077
Allantoic acid	−3.2264	0.0017108	0.0047047	1.2058
Pipecolic acid	−2.3124	0.022748	0.041705	1.2013
L-Carnitine	−2.4289	0.017903	0.034085	1.1728
Creatinine	−3.0407	0.0031613	0.0076335	1.1642
m-Aminobenzoic acid	−2.5326	0.012834	0.025931	1.1603
Indole	−2.894	0.0046922	0.010587	1.1492
L-Histidine	−2.3403	0.021428	0.040027	1.1351
3-Methylindole	−3.587	0.00051176	0.0016888	1.1249
Cinnamic acid	−2.4588	0.015506	0.030701	1.1234
Phenylacetic acid	−2.756	0.006908	0.014867	1.0948
L-Tryptophan	−2.2723	0.025179	0.044513	1.0784
Benzamide	3.0269	0.0030779	0.0076178	0.89678
5-Hydroxylysine	2.3014	0.023624	0.042523	0.84997
Pregnenolone sulfate	2.4517	0.015934	0.030931	0.79827
Sphingosine	2.2348	0.027463	0.047699	0.79163
Androsterone	3.384	0.0010576	0.0031729	0.70978
Saccharopine	3.7427	0.00029123	0.0010297	0.65817
Nutriacholic acid	4.4071	3.36 × 10^−5^	0.00014474	0.59816
Thyroxine	4.0722	9.57 × 10^−5^	0.00037908	0.59602
4-Aminohippuric acid	6.6679	2.82 × 10^−9^	2.33 × 10^−8^	0.51509
Glycyl-L-leucine	6.9782	2.37 × 10^−10^	2.34 × 10^−9^	0.49622
Vitamin D3	5.7192	1.93 × 10^−7^	1.06 × 10^−6^	0.47169
Urea	10.172	5.50 × 10^−15^	1.09 × 10^−13^	0.41933
Sphinganine	10.711	1.67 × 10^−18^	4.12 × 10^−17^	0.31225
Acetic acid	6.5618	1.11 × 10^−8^	7.85 × 10^−8^	0.2382
3-Indolepropionic acid	3.6696	0.00049507	0.0016888	0.18522
3,4,5-Trimethoxycinnamic acid	6.3521	3.30 × 10^−8^	1.92 × 10^−7^	0.1117
9-Methyluric acid	9.4247	1.56 × 10^−13^	2.58 × 10^−12^	0.059596
3-Methylxanthine	8.2614	1.96 × 10^−11^	2.42 × 10^−10^	0.026165

**Table 4 metabolites-12-00508-t004:** Statically significant metabolites among the prediabetic MetS and the normoglycemic MetS groups.

	*t*.Stat	*p*.Value	FDR	Fold Change
Threonic acid	2.7699	0.0075509	0.043973	1.9592
Indolelactic acid	3.393	0.0010926	0.0090136	1.6339
Trimethylamine	10.182	1.53 × 10^−17^	7.55 × 10^−16^	1.5493
5-Hydroxy-L-tryptophan	4.4509	2.08 × 10^−5^	0.00041226	1.5388
Quinaldic acid	2.8608	0.0055763	0.036804	1.434
Nutriacholic acid	4.105	8.81 × 10^−5^	0.001246	1.3812
Pyroglutamic acid	4.1094	7.98 × 10^−5^	0.001246	1.2941
L-Glutamic acid	2.9216	0.0042316	0.029923	1.1351
Phenylacetaldehyde	−2.9494	0.0038999	0.029699	0.82477
Urea	−6.2806	6.91 × 10^−9^	2.28 × 10^−7^	0.79831
L-Acetylcarnitine	−2.7699	0.0067091	0.041513	0.79586
5-Hydroxylysine	−3.6543	0.00040633	0.0040226	0.7697
Pregnenolone sulfate	−2.7043	0.0080906	0.044498	0.70729
2-Phenylbutyric acid	−3.9444	0.00014566	0.0018025	0.67892
Deoxycholic acid glycine conjugate	−3.5347	0.00060533	0.005448	0.54356
Vitamin D3	−5.7136	1.36 × 10^−7^	3.36 × 10^−6^	0.54128
Sphinganine	−12.621	7.37 × 10^−23^	7.29 × 10^−21^	0.27442
9-Methyluric acid	−3.7467	0.00037414	0.0040226	0.2172

## Data Availability

Data is contained within the article or Appendix A.

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
