# Peer review of "Identification of Insulin Resistance Biomarkers in Metabolic Syndrome Detected by UHPLC-ESI-QTOF-MS"

_metabolites, 2022, doi:10.3390/metabo12060508_

Round 1
Reviewer 1 Report
The study is interesting and scientifically relevant. The authors employed HPLC-ESI-QTOF to identify Diabetic Biomarkers in Metabolic Syndrome, however there are some improvements to be made.
1) Format the manuscript according to the journal's rules.
2) Has the project been approved by the ethics council? If yes, describe?
3) Analysis of metabolites is unclear. Describe in detail about the acquisition of each metabolite, its purity, among other aspects.
4) Describe in detail how the compounds were quantified in the matrix.
5) Attach chromatograms, mass spectra and chemical structures in the work.
Author Response
Dear Reviewer
Thank you for your effort in revising our manuscript.
We truly appreciate the constructive criticism and encouraging comments by the reviewers. The manuscript has been considerably revised in accordance with the comments and we believe this has significantly improved the quality of our manuscript. The revised and additional text is highlighted in yellow for quick referencing. Please find attached our responses to specific comments and point by point details.

Reviewer 2 Report
As for the scientific research, the presented data should be/can be verified by other scientists. I strongly opposed the publication without any solid and validated analytical method. In "Identification of Diabetic Biomarkers in Metabolic Syndrome Detected by HPLC-ESI-QTOF" manuscript,
- What is the data precision, repeatability and accuracy? In the supplementary, the authors should provide those data.
- RT and MS/MS were used for compounds identification, the authors should provide those data.
- How to get the fold change results, such as in table 2, 3, and 4? Do they purely based on the peak area of each analytes? Are any of the data normalized with the internal standard?
- In discussion, several referenced figure are not correct. There are no figure 3A, 3B an 3C. They should be figure 5A,, 5B and 5C.
- In this study, IDF standard was adopted for MetS. What standard was used for prediabetes? IDF or ADA?
Author Response

(The authors gave the same response as above.)

Reviewer 3 Report
The manuscript entitled “Identification of Diabetic Biomarkers in Metabolic Syndrome Detected by HPLC-ESI-QTOF- MS” studies differences in plasma metabolites between healthy, normoglycemic and prediabetic metabolic syndrome volunteers. This is an interesting paper, with important results.
My main concern is the number of the patients used in this study and whether the sample size makes the study underpowered and the fact that the three groups used are not age and BMI matched. Preferably, you should have included BMI in your analysis or used overweight individuals without any other Mets criteria as a control for comparison. Therefore, these two concerns (size and not matched population) should be addressed in the discussion as limitations of the study and pinpoint that the results must be interpreted with caution.
Some minor comments:
-Instead of “diabetic biomarkers” in the title you could refer to “insulin resistance biomarkers” as you do not use only diabetic patients
-line 20,46…MetS is a disorder characterized by a group of factors…
-line 35. The role of their alteration on some…
-line 51… please avoid the use of the word “white ethnicity”
-line 47… if you mean MetS criteria please remove “vascular inflammation”
-line 53… According to NIH
-line 62. IDF suggests central obesity and at least 2 of the 4 factors, not 3 of the 4 in general.
-line 70. Autoimmunity stands for T1D not T2D, please rephrase line 69-73.
-Line 91-105. This paragraph describing other studies should go on the discussion, otherwise if you want to leave it in the introduction please be less detailed.
-In table 1 you may use a * to show which groups have a difference in age
-In line 133 and 142 you repeat the same phrase
-In the results, when you refer to the metabolites that have different levels in different groups please do not forget to repeat which groups you compare, ie line 173 were highly downregulated in the prediabetic MetS group compared to the control group. Similarly to other comparisons as well.
-2.3 Instead of using the phrasing “pathways were impacted” you could say that you identified the metabolic pathways in which altered metabolites between groups are implicated. You cannot prove that these pathways are impacted, you have just identified in which pathways differentially expressed metabolites are taking part.
-line 269… it has “been” related
-line 279. Do you mean urea cycle?
-line 310. You may remove the sentence on phenylketonouria as this is irrelevant to your results
-You must include a paragraph with strengths and limitations of your research in the discussion
-Paragraph 4.1. You may add a sentence stating more clearly whether this study was conducted according to the guidelines of the Declaration of Helsinki and that volunteers have been informed in detail about the study before signing the consent forms.
-line 463. Instead of using the word association of metabolites with MetS, please refer to “alterations of metabolites” or “identification of differentially expressed plasma metabolites”, as you have not performed any associations in your analysis (ie regression)
-In your conclusion I think that you should rephrase the last sentence. I do not think that the likelihood of prediabetes is the result of your study, so this cannot be replicated by other studies. Do you mean that more studies are needed in larger cohorts in order to verify the implication of the above metabolites on MetS and their diagnostic value or something like that?
Author Response

(The authors gave the same response as above.)

Round 2
Reviewer 1 Report
All changes were made, but some chemical structures are not described in the supplementary material. I suggest drawing them.
Author Response
Dear Reviewer
We truly appreciate the constructive criticism and encouraging comments by the reviewers. The manuscript has been considerably revised in accordance with the comments and we believe this has significantly improved the quality of our manuscript. The revised and additional text is highlighted in yellow for quick referencing. Please find our responses to your final comment.
Comments and Suggestions for Authors
All changes were made, but some chemical structures are not described in the supplementary material. I suggest drawing them.
Response: As suggested by the reviewer, we have included the following missing chemical structures in (Supplementary supporting data S2): Glutathione, Arginine, Phenylalanine, Lysine, and Glyoxylate Dicarboxylate conjugate, please check supplementary supporting data S2.

Reviewer 2 Report
Quality of revised manuscript has been improved drastically. One final comment is,
In table S1, column 3 is measured m/z in the experiment, and column 4 is the accurate mass of each metabolite. The author needs to clarify this.
Please check the measured m/z value of 2-Phenylbutyric acid.
Author Response
Dear Reviewer
Attached please find the responses to your comments/suggestions.
We truly appreciate the constructive criticism and encouraging comments by the reviewers. The manuscript has been considerably revised in accordance with the comments and we believe this has significantly improved the quality of our manuscript. The revised and additional text is highlighted in yellow for quick referencing. Please find our responses to your final comment.
Comments and Suggestions for Authors
The quality of the revised manuscript has been improved drastically. One final comment is,
In table S1, column 3 is measured m/z in the experiment, and column 4 is the accurate mass of each metabolite. The author needs to clarify this.
Please check the measured m/z value of 2-Phenylbutyric acid.
Response: The accurate mass for 2-Phenylbutyric acid is 164.0839 and m/z measured is 147.08064
Electrospray ionization (ESI) is generally considered a soft-ionization technique, which mainly generates intact molecular ions. However, fragmentation may still happen during ionization. One common in-source fragment is the water-loss fragment [M+H-H2O]+, where a water molecule is lost during the ionization process. Moreover, adducts ions are also frequent and can occur in both, positive and negative ion modes, and some of the commonly observed additions to molecular ions as adducts include [M+H]+, [M+Na]+, [M+K]+, and [M+H-H2O]+, etc...
For 2-Phenylbutyric acid, the ions were generated in 2 types, [M+H-H2O]+ which has a high intensity, and [M+H]+ which has a low intensity. The m/z measured 147.08064 was for water-loss fragment [M+H-H2O]+
Please see the attached file
